# Coming of age in recovery: The prevalence and correlates of substance use recovery status among adolescents and emerging adults

**Douglas C. Smith**[1]*, **Crystal A. Reinhart**[1], **Shahana Begum**[1], **Janaka Kosgolla**[1], **John F. Kelly**[2], **Brandon B. Bergman**[2], **Marni Basic**[1]

**1** Center for Prevention Research and Development (CPRD), School of Social Work, University of Illinois Urbana-Champaign, Urbana, Illinois, United States of America, **2** Recovery Research Institute, Massachusetts General Hospital & Harvard Medical School, Boston, Massachusetts, United States of America

* smithdc@illnois.edu

**Data Availability Statement:** Because this study is reviewed by two different IRBs, which want assurances of how data are shared, transmitted,

## Abstract

### Background and aims

To date, no epidemiological survey has estimated the prevalence of adolescents identifying as being in recovery. This is necessary for planning and identifying the needs of youth with current and remitted substance use disorders. This study estimated the prevalence of recovery status in a large statewide epidemiological survey administered between January and March 2020.

### Participants

Participants were high school students in 9[th] through 12[th] grades throughout Illinois.

### Measurements

Youth were asked if they were in recovery and if they had resolved problems with substances. Youth who reported recovery and problem resolving dual status (DS), recovery only (RO), and problem resolution only (PRO) were compared to propensity score matched control groups who reported neither status (neither/nor; NN). Outcomes included alcohol use, binge alcohol use, cannabis use, and prescription drug use in the past 30 days.

### Findings

Prevalence estimates were 884 (1.4%) for DS, 1546 (2.5%) for PRO, and 1,811 (2.9%) for RO. Relative to propensity matched control samples, all three groups had significantly lower odds of prescription drug use. The PRO group had lower odds of past month cannabis use. There were no significant differences for either alcohol outcome.

and deidentified, we follow a data request process. Our data sharing policy can be found on https://iys.cprd.illinois.edu/results by using the link CPRD Data Sharing Policy. You may email the corresponding author (smithdc@illinois.edu) if you would like to request data.

**Funding:** The authors received support from the Illinois Department of Human Services' Division of Substance Use Prevention and Recovery (SUPR; Contract # 43CAZ03292, PI: DS) and the National Institute of Alcohol Abuse and Alcoholism (JK, K24AA022136-08 and BB, K23AA025707). The funders had no role in study design, data collection and analysis, decision to publish, or preparation of the manuscript.

**Competing interests:** The authors have declared that no competing interests exist.

## Conclusions

Prevalence estimates of youth in recovery are slightly lower than those of adults in recovery, and estimates should be replicated. Youth in recovery and those resolving problems have numerous behavioral health needs, and relative to matched controls, have even odds for past 30-day alcohol use. These findings compel us to further define recovery for adolescents and emerging adults to allow for improving treatments and epidemiological research.

## 1. Introduction

Internationally, the global burden of disease attributed to alcohol and drug use remains high [1–3]. This is especially true among adolescents and emerging adults (AEA, roughly ages 12 to 25), with alcohol and drug use representing two of the top five predictors of disability adjusted life years (DALYs) internationally. Thus, the global burden of disease attributed to alcohol and other drug use among AEA is substantial. Having a better understanding of remission and recovery processes for AEA may ultimately help mitigate this high disease burden attributable to substance use.

Although we know that 22 million adults in the United States (9.1%) report successfully resolving a problem with substances [4] and have begun to identify processes by which they achieved resolution, research with AEA is lagging. Two nationally representative surveys examined recovery among U.S. adults [4, 5]. However, according to the 2016 Surgeon General's report [6], there are multiple definitions of recovery and there is no current estimate of adolescents in recovery. This is unacceptable, as the field is at a critical juncture where there is increased emphasis on designing recovery-oriented systems of care. That is, there are increasingly more individuals identifying as recovery advocates, whose mission is to reform the substance use disorder (SUD) treatment delivery system by decreasing stigma, promoting recovery, and elevating the voices of those with lived experience [6]. Thus, we need an estimate of AEA in recovery to continue this important work. We believe our study may be the first to obtain such an estimate.

One reason why we don't have a prevalence estimate of AEA recovery is due to the lack of a standardized definition of recovery [6, 7] and reluctance to use the term recovery with adolescents [8]. Most adult definitions of recovery agree on abstinence and self-directed personal growth as two common features. However, given the lack of adolescent and emerging adult (AEA) research on recovery definitions, the United States' National Survey on Drug Use and Health (NSDUH) only asks about recovery for adults over age 18 [5]. This stems from fears that the term "recovery" will be muddied if adolescents with shorter addiction careers refer to themselves as being in recovery [8].

Research with adults suggests that recovery management services may help prevent many long-term consequences from addiction [9, 10]. It is currently unclear; however, to what extent we need such services for adolescents, and if we do, what types of services. Having better prevalence estimates and understanding more about youth who report being in recovery would be a critical first step.

### 1.1. Prior adult research on recovery

Researchers have distinguished between the overlapping concepts of problem resolution and recovery. Kelly and colleagues [11] found that approximately half of individuals who reported

resolving a significant alcohol or other drug problem considered themselves to be in recovery. Individuals in recovery experienced more severe addiction histories (i.e., prior treatment system engagement, mental health problems, and more severe and intractable problems) relative to problem resolvers. One potential implication of this finding is that individuals in recovery have had such profound experiences with substances that it became integrated into their personal self-concepts. Thus, being "in recovery" could possibly constitute one among many intersecting identities (e.g., racial, ability, or gender identities).

No studies have examined problem resolution or recovery among AEA outside of clinical samples. So, we do not understand how this population will understand the questionnaire items about resolving problems or being in recovery. Nevertheless, it is possible that some youth may experience such severe substance use that they will hit a turning point in life and consider themselves to be in recovery. Should they follow a similar pattern as adults, AEA in recovery will experience a higher prevalence of behavioral health problems than youth who resolved problems but don't consider themselves to be in recovery.

## 1.2. The present study

This paper examined two principal research questions. First, what is the prevalence of recovery and problem resolution among adolescents? Second, because we know little about youth who resolved problems or are in recovery, we compare their recent substance use to matched controls in a large epidemiological study. This permits us to consider whether problem resolving and recovering youth have lower past month substance use compared to youth with similar past year substance use. This is a good comparator for problem resolving and recovering youth, because in statewide surveys replies are typically skewed toward no substance use. We hypothesize that youth in recovery or those who have resolved problems will have lower recent substance use relative to matched controls. If past month substance use is lower among problem resolving and recovering youth versus matched controls, it would provide some initial support on the validity of problem resolution and recovery items for youth. This is because they would meet the abstinence/absence of heavy use criterion of most recovery definitions.

## 2. Method

### 2.1. Data collection

The Institutional Review Board at the lead author's university approved all the procedures for this study (Protocol #11126). Data for this study were collected through the Illinois Youth Survey (IYS), a self-report survey administered in school settings, designed to gather information about a variety of health and social indicators, including substance use patterns and attitudes of Illinois youth. The Illinois Department of Human Services (IDHS) has funded the IYS biennially since 1990.

The IYS is a school-based survey that is administered by school staff. The research team collects data from the Illinois State Board of Education on the number of schools and their enrollments by grade level. The project recruits schools by sending emails and letters to school officials, social media posts, calling schools, and having prevention specialists recruit schools. Schools send parental notifications twice prior to data collection, allowing parents to opt out their student. This study operates under a parental consent waiver. The study does not provide incentives to schools or individual students.

IYS typically yields representative statewide data that includes respondents from 8th through 12th grades using a stratified random sampling procedure. In addition to the representative statewide sample, the IYS invites all Illinois schools to participate voluntarily (i.e., volunteer sample). While not representative of the state, the volunteer sample is often substantially

larger than the random sample. Because the 2020 IYS ended prematurely during the COVID-19 pandemic and due to the anticipated low prevalence of recovery in AEA, this study uses the non-random volunteer sample. This included 610 out of 1,826 (33%) schools eligible to register for the IYS and 125,067 students. Surveys were excluded from analysis (23.8%) if participants endorsed using an attention check question such as using a fake drug, responded to less than 40% of the survey questions, or reported being dishonest. Due to the age appropriateness of the recovery and problem resolution questions [8] these were only included on the high school survey. Thus, the analytic sample for this study included 60,891 students in 9[th] through 12[th] grades.

## 2.2. Measures

**2.2.1. Recovery and problem resolution questions.**   Two survey questions were adopted from adult research studies [11] to measure problem resolution and recovery among adolescents. We measured problem resolution by asking, "*Besides nicotine, did you used to have a problem with drugs or alcohol, but no longer do*?" *(Yes/No)*. Recovery was measured by asking, "*Do you consider yourself to be in recovery*?" *(Yes/No)*. Prior to answering these questions, respondents were prompted to think about their substance use, so they would consider substance use recovery and not recovery from a mental health condition. This prompt read: "*The following questions are about RECOVERY FROM SUBSTANCE USE.*" These survey questions were then combined to create 4 groups indicating the "recovery status" of adolescents, including: 1) yes to recovery and yes to problem resolution (dual status; DS), 2) recovery only (RO), 3) problem resolution only (PRO), and 4) those saying no to both questions (neither/nor; NN).

**2.2.2. Substance use questions.**   In addition to recovery status, respondents were asked about the frequency of their substance use over the past 30 days and the past year. For both past 30 day and past year use of substances, respondents were asked "*on how many occasions (if any) have you. . .*" *(0 occasions, 1–2 occasions, 3–5 occasions, 6–9 occasions, 10–19 occasions, 20 or more occasions).*" Past 30-day use included alcohol, marijuana, and prescription drugs "not prescribed to you." Past year use included alcohol, marijuana, inhalants, MDMA/Ecstasy, LSD, cocaine, methamphetamines, heroin, prescription drugs "not prescribed to you," over-the-counter drugs, use of prescription painkillers "to get high" and use of other prescription drugs "to get high." All past year and past month substance use variables were dichotomously coded for analyses (10 + occasions = 1, < 10 occasions = 0). Additionally, respondents completed the CRAFFT [12], which is a six-item screener designed to detect adolescents with current substance use disorders (SUD). Scores of 2 or higher have high sensitivity and specificity for detecting SUD and heavy substance use [13].

**2.2.3. Behavioral health questions.**   Other survey questions used in this study covered a variety of other behavioral health problems. Two questions addressed depressed mood and suicidal ideation, including "*During the past 12 months did you ever feel so sad or hopeless almost every day for two weeks or more in a row that you stopped doing some usual activities*?" *(Yes/No)*, and "*During the past 12 months, did you ever seriously consider attempting suicide*?" *(Yes/No)*. Respondents were also asked about their gambling habits and conduct problems (i.e., getting into fights, selling drugs). A full list of questions and response choices can be found on the IYS website [14].

## 2.3. Data analyses and sample size

Among the recovery and problem resolution questions, about 25.8% of surveys had at least one missing response. The missing responses varied from 0.1% to 8.2% on other individual

survey questions. We completed Little's MCAR test [15] and results indicated that data were likely MCAR, with insignificant test results of *(# of missing patterns = 1710, chi-square = 35000.10, df = 342200, p-value = 1.00)*. We then used multiple imputation in the mice R statistical software package v3.14.0 [16] to determine the best imputation method based on parameter measurement scales. We then used predictive mean matching (PMM) for imputation and completed the imputation process two times to generate two imputed datasets. PMM uses all variables to model missing data values.

**2.3.1. Propensity score matching.**  As expected, a substantially large proportion of all high school survey respondents did not endorse being in recovery or resolving problems with substances. Among those who did not resolve problems or report being in recovery (i.e., neither/nor group), approximately half had no past year substance use. Thus, comparing these youth with low severity to those in the recovery and problem resolving groups would result in an unfair comparison of substance use outcomes in the past 30 days.

To address this problem, we utilized propensity score matching (PSM) to address confounding variables. By creating comparable groups based on confounding covariates, this approach is typically used when randomization is not an option for studying an intervention or policy [17]. We used logistic regression to estimate propensity scores that predict the likelihood of a unit obtaining the "treatment" being studied, which in our study was the likelihood of being in a recovery or problem resolving group.

Covariates were chosen based on initial bivariate analyses of potential confounding factors (See Tables 1 and 2), enhancing the model's accuracy and robustness [18]. Specifically, we used all variables in Table 2, as well as race, gender, age, geographical strata, and receipt of free and reduced priced lunch (i.e., a measure of socioeconomic status).

We used the matchit program in the R statistical package, selecting the "optimal full matching" method [19]. This approach assigns all units to a subclass, ensuring each unit gets at least one match with minimal absolute distances between control and treated groups. This technique's advantages are that it does not require an exact matching order, no units are discarded, and distances are minimized within-subclass spans. Complete matching delivers matching weights for the matched sample, making it a more suitable option for propensity score weighting [20].

After matching, we checked the standardized mean differences of covariates in the matched sample. We aimed for an absolute standardized mean difference below 0.1 for all covariates to ensure that treatment and control groups were comparable and reduce bias [18]. We selected three different matched samples, one each for the DS, RO, and PRO samples.

After successfully matching and balance checking, we performed outcome analyses on the matched sample to estimate the Average Treatment Effect on the Treated (ATT). Outcome variables included past month alcohol, cannabis, and prescription drug use, as well as binge drinking within the past two weeks. Odds ratios for all outcomes can be interpreted as the odds of reporting use in the past month or two weeks (i.e., binge alcohol use). Thus, odds ratios lower than one would indicate lower odds of recent substance use among the three recovery/problem resolution groups relative to their respective matched control samples.

## 3. Results

### 3.1. Recovery status prevalence estimates

Among all high school surveys (N = 60,891), 884 (1.4%) reported dual status (DS), 1,546 (2.5%) reported problem resolution only (PRO), and 1,811 (2.9%) reported being in recovery only (RO). Across all groups, the primary substances with which they no longer had a problem were marijuana (36.1%), alcohol (29.2%), opioids (9.9%), and other substances (16%).

**Table 1. Student characteristics among the four groups.**

| | Neither/Nor Sample | Dual Status (DS) | Recovery Only (RO) | Problem Resolution (PRO) |
|---|---|---|---|---|
| | N (%) | N (%) | N (%) | N (%) |
| **All** | **56650** | **884** | **1811** | **1546** |
| **Gender** | | | | |
| Female | 28853 (50.9) | 399 (45.1) | 864 (47.7) | 685 (44.3) |
| Male | 26820 (47.3) | 420 (47.5) | 871 (48.1) | 768 (49.7) |
| Transgender | 402 (0.7) | 31 (3.5) | 33 (1.8) | 43 (2.8) |
| Do not identify | 575 (1.0) | 34 (3.8) | 43 (2.4) | 50 (3.2) |
| **Race** | | | | |
| White | 38233 (67.5) | 552 (62.4) | 818 (45.1) | 1067 (69.0) |
| Black/African American | 3121 (5.5) | 54 (6.1) | 158 (8.7) | 101 (6.5) |
| Latino/Latina | 8555 (15.1) | 162 (18.3) | 605 (33.4) | 228 (14.7) |
| Asian American | 3722 (6.5) | 39 (4.4) | 92 (5.1) | 43 (2.8) |
| Any Other | 3019 (5.3) | 77 (8.7) | 138 (7.6) | 107 (6.9) |
| **Age (N (mean))** | 56650 (16.1) | 884 (16.3) | 1811 (16.1) | 1546 (16.3) |
| **Grade** | | | | |
| 9th | 8849 (14.9) | 86 (9.7) | 283 (15.6) | 166 (10.7) |
| 10th | 23289 (41.1) | 357 (40.4) | 795 (43.9) | 586 (37.9) |
| 11th | 7735 (13.6) | 139 (15.7) | 252 (13.9) | 223 (14.4) |
| 12th | 17177 (30.3) | 302 (34.2) | 481 (26.5) | 571 (36.9) |
| **Free/reduced lunch** | | | | |
| Free lunch | 15103 (26.7) | 340 (38.4) | 847 (46.8) | 516 (33.3) |
| Reduced price lunch | 3836 (6.8) | 69 (7.8) | 178 (9.8) | 106 (6.8) |
| Neither | 37711 (66.5) | 475 (53.8) | 786 (43.4) | 924 (59.7) |
| **Region** | | | | |
| Suburban Chicago | 40507 (71.5) | 589 (66.6) | 1269 (70.1) | 1018 (65.8) |
| Chicago | 871 (1.5) | 25 (2.8) | 68 (3.7) | 19 (1.2) |
| Other Urban | 8607 (15.1) | 158 (17.9) | 263 (14.5) | 291 (18.8) |
| Rural | 6642 (11.7) | 112 (12.7) | 211 (11.7) | 218 (14.1) |

## 3.2. Demographic characteristics

Table 1 displays adolescents' characteristics in the final analytic sample and among the recovery status groups. Relative to the neither/nor sample (50.9%), there were slightly fewer females in the DS (45.1%), RO (47.7%) and PRO (44.3%) groups. However, there was a higher percentage of transgender individuals in the DS (3.5%), RO (1.8%), and PRO (2.8%) groups relative to the neither/nor group (0.7%). Relative to the neither/nor group (15.1% Latino, 67.5% White), the RO group had a higher percentage of Latinos (33.4%) and lower percentage of Whites (45.1%). Across all groups, the average age was around 16 years of age. A majority of adolescents across all groups was from suburban Chicago. Schools in Chicago were underrepresented due to the COVID-19 pandemic.

## 3.3. Substance use and behavioral health outcomes in relation to recovery identities

Table 2 displays past year and past 30-day substance use and other behavioral health characteristics. Those in the recovery and problem resolution groups had elevated past-30 day and past year substance use relative to the neither/nor group. For practically all substances, there was a

**Table 2. Behavioral health indicators across the four groups.**

| | Neither/Nor Sample | Dual Status (DS) | Recovery Only (RO) | Problem Resolution Only (PRO) |
|---|---|---|---|---|
| | N (%) | N (%) | N (%) | N (%) |
| **All** | 56650 | 884 | 1811 | 1546 |
| **Substance use in the past 30 days** | | | | |
| Alcohol | 824 (1.5) | 125 (14.1) | 102 (5.6) | 166 (10.7) |
| Marijuana | 2555 (4.5) | 246 (27.8) | 210 (11.6) | 393 (25.4) |
| Rx drugs not prescribed | 1172 (2.1) | 221 (25.0) | 151 (8.3) | 282 (18.2) |
| **Binge drinking (2 weeks)** | 591 (1.0) | 103 (11.6) | 89 (4.9) | 157 (10.1) |
| **Substance use in the past year** | | | | |
| Alcohol | 5204 (9.2) | 325 (36.8) | 246 (14.5) | 367 (23.7) |
| Marijuana | 4741 (8.4) | 404 (45.7) | 323 (17.8) | 703 (45.4) |
| Inhalants | 407 (0.7) | 102 (11.5) | 62 (3.4) | 125 (8.1) |
| MDMA, Ecstasy | 307 (0.5) | 92 (10.4) | 62 (3.4) | 130 (8.4) |
| LSD | 349 (0.6) | 96 (10.9) | 66 (3.0) | 146 (9.4) |
| Cocaine | 342 (0.6) | 100 (11.3) | 67 (3.7) | 128 (8.2) |
| Methamphetamines | 297 (0.5) | 95 (10.7) | 59 (3.2) | 121 (7.8) |
| Heroin | 319 (0.5) | 89 (10.0) | 58 (3.2) | 113 (7.3) |
| Rx drugs not prescribed | 1891(3.3) | 305 (34.5) | 187 (10.3) | 399 (25.8) |
| Over the counter drugs | 407 (0.5) | 141 (15.9) | 80 (4.4) | 176 (11.4) |
| Rx painkillers to get high | 362 (0.6) | 144 (16.3) | 77 (4.2) | 195 (12.6) |
| Other Rx drugs to get high | 665 (11.9) | 216 (24.4) | 101 (5.6) | 267 (17.3) |
| **Mental Health and CRAFFT** | | | | |
| Sadness/Hopelessness 2+weeks | 18581 (32.8) | 219 (24.8) | 974 (53.7) | 964 (61.1) |
| Suicidal ideation | 7772 (13.6) | 381 (43.0) | 515 (28.4) | 589 (38.0) |
| 2+ CRAFFT score | 9378 (16.5) | 141 (15.9) | 768 (42.4) | 413 (26.7) |
| **Gambling** | | | | |
| Gambled via machine | 1518 (2.7) | 165 (18.7) | 147 (8.1) | 213 (13.8) |
| Gambled via online | 2148 (3.8) | 187 (21.2) | 179 (9.9) | 240 (15.5) |
| Felt bad about gambling | 2135 (3.8) | 208 (23.5) | 220 (12.1) | 223 (14.4) |
| Gamble more than planned | 1655 (2.9) | 217 (24.5) | 160 (8.8) | 226 (14.6) |
| **Conduct Problems** | | | | |
| Physical fight | 8052 (14.2) | 183 (20.7) | 548 (30.2) | 621 (40.1) |
| Carried weapon | 4711 (8.3) | 195 (22.0) | 320 (17.7) | 487 (31.5) |
| Sold illegal drugs | 1713 (3.0) | 197 (22.2) | 228 (12.5) | 449 (29.0) |
| Been drunk or high | 4224 (7.4) | 365 (41.3) | 472 (26.1) | 738 (47.7) |

consistent rank order of highest to lowest prevalence of substance use with DS being highest and followed by, in descending order, PRO, RO, and the NN groups. Regarding probable substance use disorders, the DS group had the lowest prevalence (15.9%) of meeting the threshold on the CRAFFT screener, followed by the NN (16.5%), PRO (26.7%), and RO groups (42.4%). Substance use severity was much higher for youth in the recovery/problem resolution groups relative to the rest of the sample.

Regarding mental health functioning, gambling, and conduct behaviors, a similar pattern emerged. With few exceptions, all recovery and problem resolution groups had elevated risks relative to the neither nor group (NN). Those in the DS group had the highest prevalence followed by those in the PRO, RO, and NN groups. The only exception to this pattern was that DS youth had slightly lower prevalence of past year sadness/hopelessness relative to the other

groups. However, the DS group had an elevated prevalence of suicidal ideation (43.0%) relative to the RO (28.4%), PRO (38.0%), and NN (13.6%) groups.

### 3.4. Bias reduction and ATT analyses

Based on the findings of Figs 1–3, propensity matching resulted in a significant decrease in bias between the recovery/problem resolving groups and their respective matched control groups. The absolute standardized mean difference for most confounding factors was below 0.1. While a few factors did show a slightly higher absolute standardized mean difference, our propensity score matching method reduced bias considerably.

The ATT plots are given in Figs 4–6. Fig 4 shows that youth in the RO group were significantly less likely to use prescription drugs relative to matched controls (*OR = 0.74, 95% CI [0.63, 0.86]*). Being in the RO group was associated with a 26% reduction in the odds of prescription drug misuse in the past 30 days. Similarly, Figs 5 and 6 show that participants in the PRO (*OR = 0.76; 95% CI [0.64, 0.92]*) and DS groups (*OR = 0.76, 95% CI [0.64, 0.92]*) also had lower odds of past 30 day prescription drug misuse.

Regarding marijuana use, youth in the PRO (*OR = 0.88, [0.78, 1.00] p < .05*) had lower odds of use relative to matched controls. Although non-significant, the DS group also had lower odds of past 30-day cannabis use (*OR = 0.86, [0.72, 1.03], p < .10*). This equated to a 12% and 14% reduction in the odds of past month marijuana use, respectively.

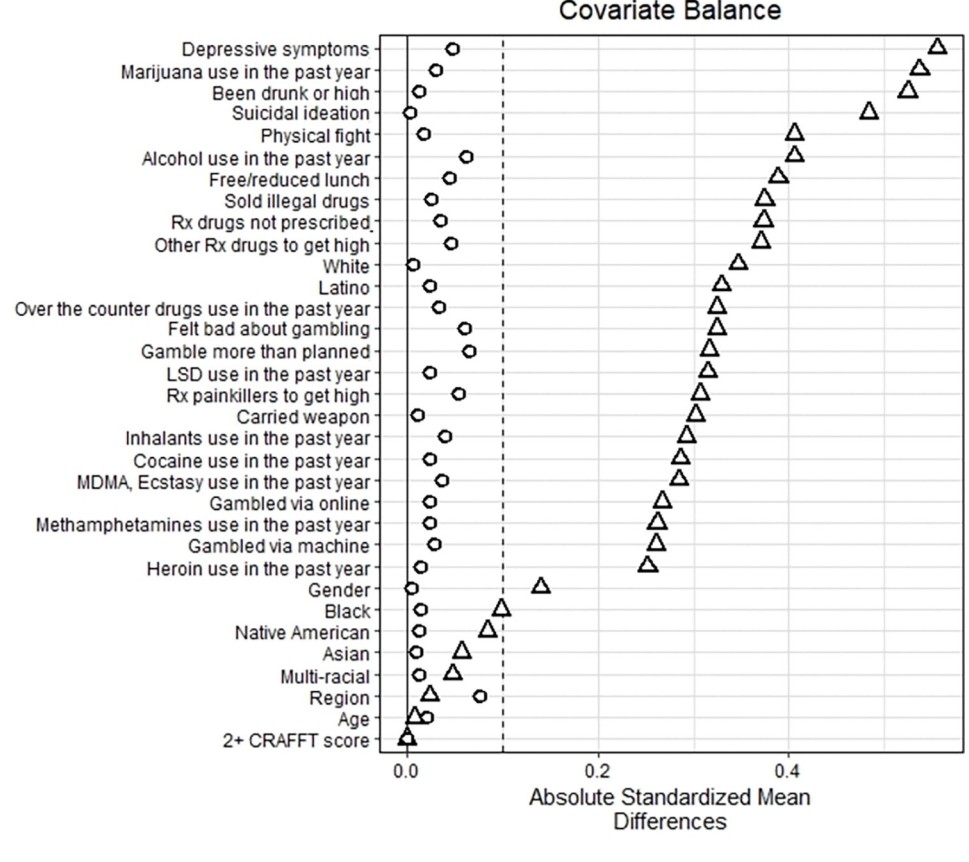

**Fig 1. Bias reduction achieved for youth in Recovery Only (RO).**

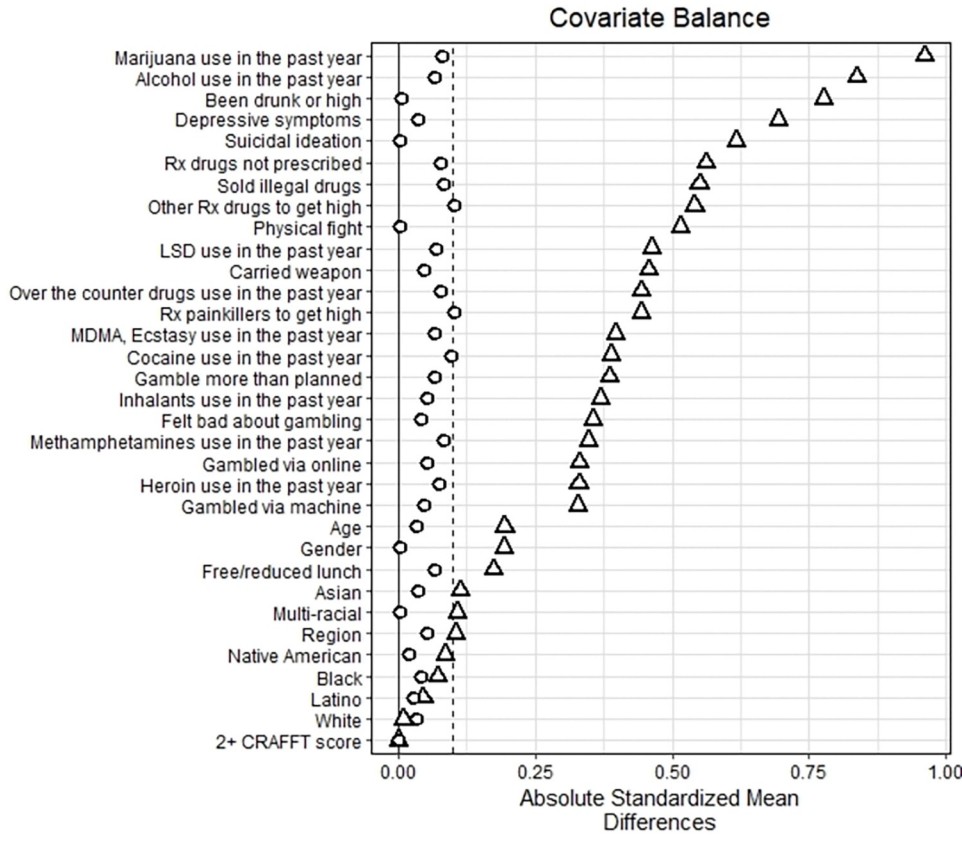

**Fig 2. Bias reduction achieved for youth in Problem Resolution Only (PRO).**

Regarding alcohol use, none of the three recovery or problem resolution groups had significantly different odds of past month alcohol use. Similarly, none of these groups had significantly different binge alcohol use in the past two-week period.

## 4. Discussion

### 4.1. Adolescent recovery prevalence estimate

This is the first large epidemiological survey to estimate the percentage of adolescents considering themselves in recovery, which was 4.3% when the dual status (DS) and recovery only (RO) groups were pooled. In 2020, there were approximately 303,000 adolescents in 10th and 12th grade across all of Illinois [21]. Thus, if we replicate this in a representative survey, it would equate to approximately 13,029 students in these grades across Illinois who consider themselves in recovery. As adolescents in high school likely have shorter substance use careers, it seems intuitive that our prevalence estimate falls within the lower range of estimates for adults. Population estimates of adult recovery range from 4.1% [4] to 8.3% [5] in the two existing national studies.

**4.1.1. Validity of the concept of recovery for adolescents.** Although much more work needs to be done on how youth understand our survey item on recovery status, our findings potentially validate the use of this item with this population. As seen in Table 2, youth in the

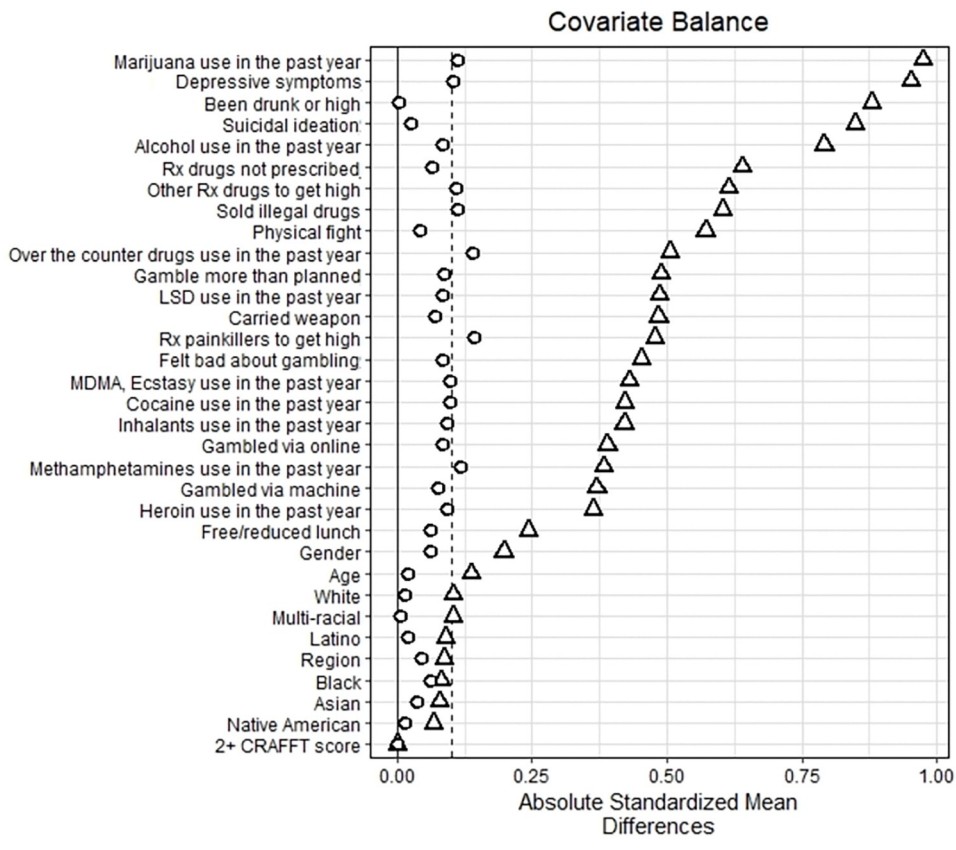

**Fig 3. Bias reduction achieved for youth in Dual Status (DS).**

DS and RO groups typically had elevated (past month) substance use and mental health problems relative to the PRO and NN groups. This supports the possibility that these youth are "in recovery," as our results are consistent with adult research showing that those in recovery had

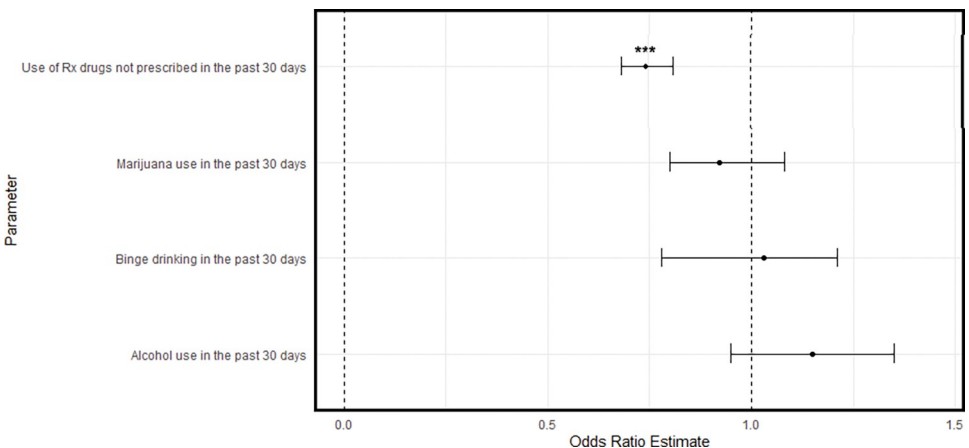

**Fig 4. Comparison of past 30-day substance use between RO group and matched controls.**

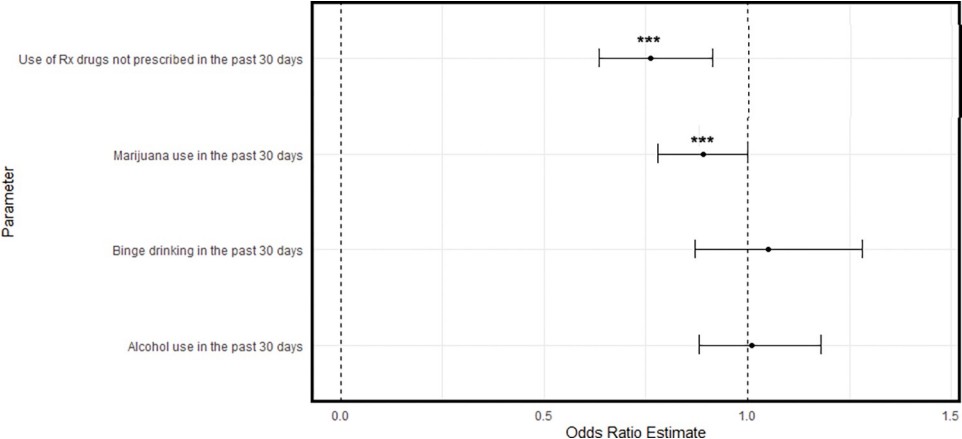

**Fig 5. Comparison of past 30-day substance use between PRO group and matched controls.**

more severe substance use than those with problem resolution only [11]. In the current study, both groups were more severe than youth not in recovery or experiencing problem resolution.

We also saw in our descriptive analyses that CRAFFT scores, a measure of probable SUD in the past 12 months, were lower for youth in the DS group than all other groups. This could be interpreted as indicating that this group is succeeding in achieving remission from substance related problems. Conceptually, it makes sense that a higher proportion of youth who have not resolved problems but are in in recovery (RO) were still exceeding clinical thresholds for probable SUD (42.4%) relative to those who said they were in recovery and had resolved problems (15.9%). A high proportion of these DS youth (about 84%) could arguably be in remission, as they were at comparable levels of meeting the CRAFFT threshold relative to the neither/nor group (16.5%, or 83.5% as subthreshold for probable SUD). However, because our data are cross sectional, this remains open to interpretation.

The comparisons of past month use between the three recovery/problem resolution groups are difficult to interpret. It is encouraging that all three groups had lower odds of prescription drug misuse than their propensity matched control samples, and that the odds of past month marijuana use were significantly lower for the PRO group and trending in that direction for the DS group ($p < .10$). However, alcohol use odds were similar between all recovery/problem

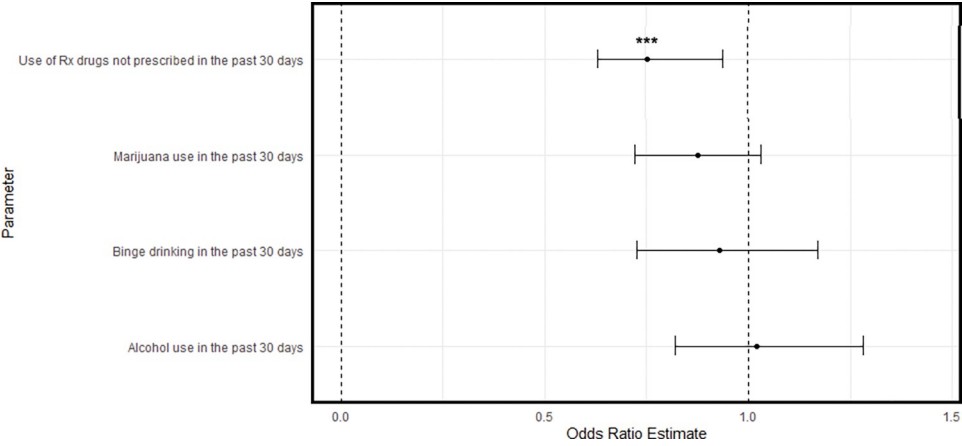

**Fig 6. Comparison of past 30-day substance use between DS group and matched controls.**

resolution groups and their respective matched control groups. This indicates the continued need for interventions targeting risky drinking practices among problem resolving and recovering youth. Although they may have made substantial changes with regards to other substances, their odds of drinking on ten or more occasions in the past month were comparable to matched controls.

Because we do not know if these adolescents endorse abstinence goals, alternative possibilities could be considered. First, youth may be having success with the substance they had the most trouble with and are able to use other non-problematic substances moderately. This could explain why all groups had similar past month alcohol use relative to their control groups. Additional research should explore patterns of non-abstinent recovery among youth, which is quite common among adults with less severe substance use problem histories [22]. However, we also note that approximately one in ten youth in both the DS and RO groups reported recent binge drinking. Furthermore, youth in clinical samples do not always define recovery as involving complete abstinence [8]. More work should explore whether youth in recovery continue to use. Another possibility is that some youth in these groups are working on abstinence-based recovery and have relapsed within the past month. Future research should ask youth about their definition of recovery, and if it includes moderate use of substances. Additionally, future research surveys could ask youth directly if they considered recent use to constitute a relapse.

## 4.2. Behavioral health needs of problem resolving and recovering adolescents

The DS, PRO, and RO youth had multiple behavioral health comorbidities. These findings are consistent with studies of adolescents in treatment who have extensive behavioral health needs such as those observed here [23]. They point to the need to assess and treat co-occurring conditions among youth who are in recovery or report having resolved a problem with substances.

## 4.3. Study implications and future research recommendations

Future research should replicate these prevalence estimates using representative sampling. If the estimate here is accurate, this would have large implications for planning what types of recovery supports are needed for such youth. However, it may be that because of prior concerns about using the term recovery with adolescents, more preliminary steps are needed to work toward increasing researchers' confidence in estimating AEA recovery in larger epidemiological studies.

Although we found here that recovery status was associated with higher substance use and behavioral health needs, some conceptual work may be needed regarding what adolescents and emerging adults are thinking when they respond to recovery items on a survey. Thus, cognitive interviewing with youth responding to the item "Do you consider yourself to be in recovery?" is recommended. Cognitive interviewing involves assessing participants' understanding of survey items and has successfully identified developmental differences in AEAs' conceptualizations of substance use disorder symptoms. For example, Slade and colleagues found that young people did not interpret the diagnostic criterion of using for longer than intended as indicating a loss of control over substances [24]. In our context, it could be that young people are interpreting the words "recovery" or "problem resolution" in ways inconsistent with existing definitions. Thus, carefully probing on why youth responded "yes" to these items may yield important insights. For example, such research could identify the potential for false positives in epidemiological estimates of youth recovery should these items be considered for use nationally.

Similarly, future studies on youth in recovery should take detailed substance use histories to confirm that youth in recovery had a prior substance use disorder (SUD). Although we administered the CRAFFT in this study, we did not have detailed or temporally lagged SUD variables available for analysis. So, given the accepted definition of a prior SUD as a prerequisite for being in recovery [6, 7], it is critical to know if the prevalence estimate here may be accurate. Although our data show associations between recovery status and substance use indicators, one disturbing possibility is that some AEA with very limited substance use histories are endorsing that they are in recovery. In our study we had no way of knowing if abstinent youth reporting recovery were successful in their recovery efforts, or if they were youth who misunderstood the question and had limited substance use histories.

In addition to resolving these issues surrounding obtaining accurate prevalence estimates of AEA in recovery, we recommend more work on how AEA define their recovery. In our exploratory analyses, we saw substance use patterns that don't conform to the traditional abstinence-based definition of recovery for AEA [25]. In fact, a scoping review of recovery narratives among people with alcohol use disorders revealed little AEA research and less research on non-abstinent recovery [26]. It seems likely that there are developmental differences in recovery processes, and future work should illuminate these processes specifically for AEA. Importantly, such work should be done with a representative sample of AEA in recovery, and not just AEA in clinical settings. This is because only a small number of people with substance use disorders may need treatments to successfully remit [27].

Thus, one intriguing line of inquiry is whether AEA can successfully maintain non-abstinent recovery, and how this may be related to future health outcomes. Latent profile analyses of adults three years after alcohol use disorder treatment identified a group of high functioning heavy drinkers [28]. AEA research is missing, especially among diverse samples of community-dwelling young people who may exhibit even more varied patterns of recovery versus those identified in treatment research.

## 4.4. Limitations

We caution readers to interpret this study's findings in light of its limitations. First, the generalizability of this survey is limited. Although we planned to complete a statewide representative survey, the global COVID-19 pandemic interrupted our study. Thus, although this is to our knowledge the largest study estimating adolescent and emerging adult (AEA) recovery statuses, it may not generalize to all high school youth in Illinois or the United States. The city of Chicago was underrepresented in this survey. Chicago schools historically survey after the month of March, after when COVID-19 prematurely ended the study in 2020. Additionally, the survey may have not sampled youth who had severe substance use and were not present for participating in a school-based survey. Second, we must remain open to the possibility that self-report was inaccurate. As we noted, participants' understanding of our recovery item may be very different from generally accepted definitions of substance use recovery. Furthermore, lack of attention, carelessness, or mischievous responding could also possibly account for our findings, as willful misrepresentation of rarely endorsed items is expected to have a greater impact on items with low base rates [29] such as recovery identities. Finally, we note that our propensity matching procedures may have influenced the findings, as past year and past month substance use are correlated in this cross-sectional study. Yet, we argue that it is important to match youth on past year substance use to have a fair comparator for recovering and problem resolving youth. That is, it would also be problematic to compare recovering and problem resolving youth to participants in the sample with low levels of substance use (see neither/nor column in Table 2) had we just matched on demographics. Future research should address these limitations.

## 5. Conclusion

This analysis of the 2020 Illinois Youth Survey data estimates that 4.3% of youth in 9th through 12th grade consider themselves to be in recovery. An additional 2.5% consider themselves to have resolved a problem with substances, yet do not consider themselves to be in recovery. Notwithstanding unresolved limitations and conceptual issues, if these are accurate estimates, they have large implications for medical professionals designing recovery supports for AEA. These findings, and our literature review discovering very little AEA recovery research, also suggest a pressing need for more research on community-dwelling youth in recovery. Such research may illuminate what helped them resolve their substance use problems, their conceptual understanding of, and perceptions regarding, the term "recovery," and the meaning of identifying as a person "in recovery." Such investigation may help ultimately uncover and identify the kinds of approaches that may be best suited to attracting and engaging youth earlier in the course of experiencing a substance use disorder. Finally, it is critical to understand how youth in recovery and problem resolvers sustain these salubrious changes over time. In turn, this knowledge could help alleviate the considerable national public health, public safety, and economic burdens associated with a more protracted course of SUD.

## Author Contributions

**Conceptualization:** Douglas C. Smith.

**Formal analysis:** Shahana Begum, Janaka Kosgolla.

**Funding acquisition:** Douglas C. Smith, Crystal A. Reinhart.

**Methodology:** Shahana Begum.

**Project administration:** Douglas C. Smith, Marni Basic.

**Supervision:** Douglas C. Smith.

**Writing – original draft:** Douglas C. Smith, Crystal A. Reinhart, Shahana Begum, Janaka Kosgolla, Brandon B. Bergman.

**Writing – review & editing:** Douglas C. Smith, Crystal A. Reinhart, Shahana Begum, Janaka Kosgolla, John F. Kelly, Brandon B. Bergman, Marni Basic.

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
