## [Decision Letter · Decision Letter 0]

12 May 2023

PONE-D-23-08858Coming of Age in Recovery: The prevalence and correlates of substance use recovery status among adolescents and emerging adultsPLOS ONE

Dear Dr. Smith,

Thank you for submitting your manuscript to PLOS ONE. After careful consideration, we feel that it has merit but does not fully meet PLOS ONE’s publication criteria as it currently stands. Therefore, we invite you to submit a revised version of the manuscript that addresses the points raised during the review process.

The manuscript presents important contributions to public health. However, the reviewers raised important concerns during the evaluation process that should be addressed by the authors, particularly the second reviewer. In order to add value to the manuscript, the authors should revise the seven concerns listed below:

The first concern is that the authors used recovery status as the outcome when it should be the exposure.

The second concern is that the authors should use methods such as propensity score matching or weighting to make the exposed and unexposed groups more comparable.

The third concern is that the authors included all exposures in the regression models simultaneously, which could lead to spurious associations.

The fourth concern is that individuals with severe substance use disorders may not participate in school-based surveys, leading to potential limitations.

The fifth concern is that the authors did not specify what variables were used in the imputation model.

The sixth concern is that the authors should provide more concrete steps on future development of the items on recovery from substance use in the population of adolescents and emerging adults.

The seventh concern is that the definitions/conceptualization of problem resolution and recovery were not clear in the methods.

We look forward to receiving your revised manuscript.

Kind regards,

Ricardo de Mattos Russo Rafael, Ph.D.

Academic Editor

PLOS ONE

Journal Requirements:

Reviewers' comments:

Reviewer's Responses to Questions

**Comments to the Author**

1. Is the manuscript technically sound, and do the data support the conclusions?

Reviewer #1: Yes

Reviewer #2: Partly

2. Has the statistical analysis been performed appropriately and rigorously? 

Reviewer #1: Yes

Reviewer #2: Yes

3. Have the authors made all data underlying the findings in their manuscript fully available?

Reviewer #1: Yes

Reviewer #2: No

4. Is the manuscript presented in an intelligible fashion and written in standard English?

Reviewer #1: Yes

Reviewer #2: Yes

5. Review Comments to the Author

Reviewer #1: This is a well written paper on an important public health problem. The authors set out to describe the pattern of recovery among adolescents and youth in the US. The methods are well described and the findings well presented. The discussion is balanced and considers the study limitations. I have one minor comment: The definitions/conceptualization of problem resolution and recovery did not come out well in the methods. The authors should provide a description of these terms to improve understandability.

Reviewer #2: In the present study, the authors examined the prevalence of recovery and problem resolution in a sample of adolescents and young adults and the sociodemographic and behavioral characteristics associated with these. The study is well written and addresses an important yet understudied problem, however, I have several major concerns that would require further attention from the authors.

1. The authors used the categorical variable recovery status as the outcome, however, I believe that the recovery status is supposed to be the exposure in this study. The hypothesis listed in introduction “given prior research, we hypothesized that youth in recovery would have more severe substance use and behavioral health problems relative to those not in recovery” supports the idea that the exposure is indeed recovery status and the outcome the sociodemographic and behavioral characteristics. This direction is further supported by claims made in the abstract “chi square, ANOVA, and regression analyses compared demographic variables, past 30 day and past year alcohol and other drug use, and behavioral outcomes among four groups of youth”, methods “use of any substance was used as the reference category for outcomes”, and results “3.3. Substance Use and Behavioral Health Outcomes in Relation to Recovery Identities”.

Moreover, the authors randomly selected a subset of individuals who responded no to both questions on recovery and problem resolution (NN group) but had used at least one substance in the past year, and used this subset of participants as the comparison group in their analyses. They proclaimed that “the random selection process successfully balanced demographics between the selected NN group and the three other recovery status groups”. The descriptive statistics in Table 1 indicate that this group differed from the other groups on most of the assessed characteristics. However, since recovery status was used as the outcome in the present version of the study, there should be a balanced covariate distribution between the exposed and unexposed group (e.g., individuals with 2+ CRAFFT score and individuals with lower CRAFT score). This also suggests that recovery status should be the exposure and sociodemographic and behavioral characteristics the outcome.

While the reversal of exposure and outcome is generally discouraged since these should be, ideally, defined in the study protocol, I believe that the authors actually intended to use recovery status as the exposure and the sociodemographic and behavioral characteristics as the outcome. The authors should clarify what was the direction of the association that they intended to study. If indeed the outcome of interest were the sociodemographic and behavioral characteristics, then they should reverse the exposure-outcome pairs.

2. Regardless of what was the intended exposure and outcome, to increase the confidence that the observed effects are due to intrinsic differences between the exposed and unexposed group and not due to differences between these on covariates (ie, unaccounted confounding), the authors could utilize several methods and techniques that would make the exposed and unexposed group more comparable. Popular options that the authors could consider are propensity score matching (e.g., 10.1093/oxfordjournals.aje.a010011) or propensity score weighting (e.g., 10.1136/bmj.l5657).

3. Putting aside whether these were the actual exposures, it seems to me that the authors included all exposures (i.e, sociodemographic and behavioral characteristics) in the regression models simultaneously, as indicated by Table 3 and 4. Such an approach would be problematic because it is likely that there are complex relationships between the included exposures: these could lead to spurious associations, including the reversal of the direction of the associations (e.g. 10.1097/EDE.0000000000001259, 10.1016/j.jclinepi.2022.05.021).

The authors should clarify whether they included all exposures in the models simultaneously. If so, then they should fit models containing the confounders and each of the exposure separately.

4. I have no experiences with school-based surveys conducted in the United States, but it is a known limitation of general population surveys that people with substance use disorders, in particular those with severe ones, are substantially less likely to participate in them.

Is it possible that individuals who would have the most severe substance use disorders would not even be present at school, therefore would be unreachable by this survey? Complementarily, is it possible that these individuals would be less likely to participate in the survey when present in school? The authors should clarify these and expand the limitations section accordingly.

5. The authors indicated that they used multiple imputation to impute the missing data, however, it is not clear what was present in the imputation model. Thus, the authors should elaborate what variables were used in the imputation model.

6. I believe that the authors should provide more concrete steps on future development of the items on recovery from substance use in the population of adolescents and emerging adults. In particular, to ensure that each member of the target population has the same understanding of the underlying concept (i.e., recovery), the authors could provide potential directions for developing a definition of recovery in the context of adolescents and emerging adults. This would, then, ensure that the responses refer to the same concept and not multiple, potentially incomparable concepts.

6. PLOS authors have the option to publish the peer review history of their article (what does this mean?). If published, this will include your full peer review and any attached files.

Reviewer #1: **Yes: **Florence Jaguga

Reviewer #2: No

---

## [Author Response · Author response to Decision Letter 0]

10 Aug 2023

Reviewer Comments Author Responses

Reviewer #1: This is a well written paper on an important public health problem. The authors set out to describe the pattern of recovery among adolescents and youth in the US. The methods are well described and the findings well presented. The discussion is balanced and considers the study limitations. 

I have one minor comment: The definitions/conceptualization of problem resolution and recovery did not come out well in the methods. The authors should provide a description of these terms to improve understandability. 

We appreciate the kind comments on the paper’s novel contributions to the field. 

We have tried to improve the clarity of the problem resolution and recovery terms by revising the section of the introduction entitled “Recovery Research with Adults.”

However, we also note that there is very limited work on defining these terms for adolescents. So, one of the main recommendations we have in the discussion section is to do more research on defining these terms. 

Reviewer #2: In the present study, the authors examined the prevalence of recovery and problem resolution in a sample of adolescents and young adults and the sociodemographic and behavioral characteristics associated with these. The study is well written and addresses an important yet understudied problem, however, I have several major concerns that would require further attention from the authors.

We appreciate the reviewer’s recognition that the study makes a novel contribution in an understudied area. We have worked hard to be as responsive to the methodological and conceptual comments made by Reviewer 2.

1. The authors used the categorical variable recovery status as the outcome, however, I believe that the recovery status is supposed to be the exposure in this study. The hypothesis listed in introduction “given prior research, we hypothesized that youth in recovery would have more severe substance use and behavioral health problems relative to those not in recovery” supports the idea that the exposure is indeed recovery status and the outcome the sociodemographic and behavioral characteristics. This direction is further supported by claims made in the abstract “chi square, ANOVA, and regression analyses compared demographic variables, past 30 day and past year alcohol and other drug use, and behavioral outcomes among four groups of youth”, methods “use of any substance was used as the reference category for outcomes”, and results “3.3. Substance Use and Behavioral Health Outcomes in Relation to Recovery Identities”. 

Moreover, the authors randomly selected a subset of individuals who responded no to both questions on recovery and problem resolution (NN group) but had used at least one substance in the past year, and used this subset of participants as the comparison group in their analyses. They proclaimed that “the random selection process successfully balanced demographics between the selected NN group and the three other recovery status groups”. The descriptive statistics in Table 1 indicate that this group differed from the other groups on most of the assessed characteristics. However, since recovery status was used as the outcome in the present version of the study, there should be a balanced covariate distribution between the exposed and unexposed group (e.g., individuals with 2+ CRAFFT score and individuals with lower CRAFT score). This also suggests that recovery status should be the exposure and sociodemographic and behavioral characteristics the outcome.

 We have followed this reviewer’s suggestion to use propensity matching, which addresses this comment. We feel the resulting manuscript is much stronger.

While the reversal of exposure and outcome is generally discouraged since these should be, ideally, defined in the study protocol, I believe that the authors actually intended to use recovery status as the exposure and the sociodemographic and behavioral characteristics as the outcome. The authors should clarify what was the direction of the association that they intended to study. If indeed the outcome of interest were the sociodemographic and behavioral characteristics, then they should reverse the exposure-outcome pairs. 

As noted above, we did this. Thank you for the suggestion.

2. Regardless of what was the intended exposure and outcome, to increase the confidence that the observed effects are due to intrinsic differences between the exposed and unexposed group and not due to differences between these on covariates (ie, unaccounted confounding), the authors could utilize several methods and techniques that would make the exposed and unexposed group more comparable. Popular options that the authors could consider are propensity score matching (e.g., 10.1093/oxfordjournals.aje.a010011) or propensity score weighting (e.g., 10.1136/bmj.l5657). 

We thank the reviewer for this excellent suggestion. Using optimal full matching, a form of propensity matching, we balanced covariates on (measured) potentially confounding variables.

3. Putting aside whether these were the actual exposures, it seems to me that the authors included all exposures (i.e, sociodemographic and behavioral characteristics) in the regression models simultaneously, as indicated by Table 3 and 4. Such an approach would be problematic because it is likely that there are complex relationships between the included exposures: these could lead to spurious associations, including the reversal of the direction of the associations (e.g. 10.1097/EDE.0000000000001259, 10.1016/j.jclinepi.2022.05.021). 

In this major revision, all confounds are accounted for in the match. Thus, the only variables in the models are the exposures (recovery groups) and outcomes (past 30 day substance use and binge drinking in the past two weeks.)

The authors should clarify whether they included all exposures in the models simultaneously. If so, then they should fit models containing the confounders and each of the exposure separately. 

Optimal full matching addresses all the potential confounders in this analysis. 

4. I have no experiences with school-based surveys conducted in the United States, but it is a known limitation of general population surveys that people with substance use disorders, in particular those with severe ones, are substantially less likely to participate in them. 

We appreciate this comment. We have added a sentence in the limitations section about this.

Is it possible that individuals who would have the most severe substance use disorders would not even be present at school, therefore would be unreachable by this survey? Complementarily, is it possible that these individuals would be less likely to participate in the survey when present in school? The authors should clarify these and expand the limitations section accordingly. 

We appreciate this comment. We have added a sentence in the limitations section about this.

5. The authors indicated that they used multiple imputation to impute the missing data, however, it is not clear what was present in the imputation model. Thus, the authors should elaborate what variables were used in the imputation model.

 In this version of the manuscript, we clarified the imputation method (predictive mean matching in the R package, mice). This uses all variables in the dataset.

6. I believe that the authors should provide more concrete steps on future development of the items on recovery from substance use in the population of adolescents and emerging adults. In particular, to ensure that each member of the target population has the same understanding of the underlying concept (i.e., recovery), the authors could provide potential directions for developing a definition of recovery in the context of adolescents and emerging adults. This would, then, ensure that the responses refer to the same concept and not multiple, potentially incomparable concepts. 

We agree, and have proposed to use cognitive interviewing methods to check their understanding of these items. This would be a logical next step for item and scale development.

---

## [Decision Letter · Decision Letter 1]

20 Sep 2023

PONE-D-23-08858R1Coming of Age in Recovery: The prevalence and correlates of substance use recovery status among adolescents and emerging adultsPLOS ONE

Dear Dr. Smith,

Thank you for submitting your manuscript to PLOS ONE. After careful consideration, we feel that it has merit but does not fully meet PLOS ONE’s publication criteria as it currently stands. Therefore, we invite you to submit a revised version of the manuscript that addresses the points raised during the review process.

The authors have made partial improvements in response to the reviewers' comments. We appreciate their efforts in this regard. However, there are still unresolved issues concerning the analysis, as pointed out by the reviewer. Therefore, we respectfully request that these issues be addressed.

In order to assist the authors, the specific items referred to in the review are:

“1. The authors used the categorical variable recovery status as the outcome, however, I believe that the recovery status is supposed to be the exposure in this study. The hypothesis listed in introduction “given prior research, we hypothesized that youth in recovery would have more severe substance use and behavioral health problems relative to those not in recovery” supports the idea that the exposure is indeed recovery status and the outcome the sociodemographic and behavioral characteristics. This direction is further supported by claims made in the abstract “chi square, ANOVA, and regression analyses compared demographic variables, past 30 day and past year alcohol and other drug use, and behavioral outcomes among four groups of youth”, methods “use of any substance was used as the reference category for outcomes”, and results “3.3. Substance Use and Behavioral Health Outcomes in Relation to Recovery Identities”.

Moreover, the authors randomly selected a subset of individuals who responded no to both questions on recovery and problem resolution (NN group) but had used at least one substance in the past year, and used this subset of participants as the comparison group in their analyses. They proclaimed that “the random selection process successfully balanced demographics between the selected NN group and the three other recovery status groups”. The descriptive statistics in Table 1 indicate that this group differed from the other groups on most of the assessed characteristics. However, since recovery status was used as the outcome in the present version of the study, there should be a balanced covariate distribution between the exposed and unexposed group (e.g., individuals with 2+ CRAFFT score and individuals with lower CRAFT score). This also suggests that recovery status should be the exposure and sociodemographic and behavioral characteristics the outcome.

While the reversal of exposure and outcome is generally discouraged since these should be, ideally, defined in the study protocol, I believe that the authors actually intended to use recovery status as the exposure and the sociodemographic and behavioral characteristics as the outcome. The authors should clarify what was the direction of the association that they intended to study. If indeed the outcome of interest were the sociodemographic and behavioral characteristics, then they should reverse the exposure-outcome pairs.

2. Regardless of what was the intended exposure and outcome, to increase the confidence that the observed effects are due to intrinsic differences between the exposed and unexposed group and not due to differences between these on covariates (ie, unaccounted confounding), the authors could utilize several methods and techniques that would make the exposed and unexposed group more comparable. Popular options that the authors could consider are propensity score matching (e.g., 10.1093/oxfordjournals.aje.a010011) or propensity score weighting (e.g., 10.1136/bmj.l5657).

3. Putting aside whether these were the actual exposures, it seems to me that the authors included all exposures (i.e, sociodemographic and behavioral characteristics) in the regression models simultaneously, as indicated by Table 3 and 4. Such an approach would be problematic because it is likely that there are complex relationships between the included exposures: these could lead to spurious associations, including the reversal of the direction of the associations (e.g. 10.1097/EDE.0000000000001259, 10.1016/j.jclinepi.2022.05.021).

The authors should clarify whether they included all exposures in the models simultaneously. If so, then they should fit models containing the confounders and each of the exposure separately.”

We look forward to receiving your revised manuscript.

Kind regards,

Ricardo de Mattos Russo Rafael, Ph.D.

Academic Editor

PLOS ONE

Reviewers' comments:

Reviewer's Responses to Questions

**Comments to the Author**

1. If the authors have adequately addressed your comments raised in a previous round of review and you feel that this manuscript is now acceptable for publication, you may indicate that here to bypass the “Comments to the Author” section, enter your conflict of interest statement in the “Confidential to Editor” section, and submit your "Accept" recommendation.

Reviewer #2: (No Response)

2. Is the manuscript technically sound, and do the data support the conclusions?

Reviewer #2: Partly

3. Has the statistical analysis been performed appropriately and rigorously? 

Reviewer #2: No

4. Have the authors made all data underlying the findings in their manuscript fully available?

Reviewer #2: No

5. Is the manuscript presented in an intelligible fashion and written in standard English?

Reviewer #2: Yes

6. Review Comments to the Author

Reviewer #2: 1. to 3. Thank you for addressing my points. Using propensity score matching reduced the covariate imbalance between the groups, however, the authors performed matching also on past year substance use and CRAFT score. Since the outcome in models is 30-day substance use, by matching on covariates that are proxy for substance use, the authors de facto matched on the outcome. This can be responsible for detecting mostly null effects in the regression models. Thus, the authors should rerun the matching without matching on substance use and CRAFT score.

4. to 6. These points were addressed sufficiently by the authors. Thank you.

7. PLOS authors have the option to publish the peer review history of their article (what does this mean?). If published, this will include your full peer review and any attached files.

Reviewer #2: No

---

## [Author Response · Author response to Decision Letter 1]

31 Oct 2023

Please see our attached cover letter for our response to the last reviewer comment. Per our last correspondence with the journal, this was how you wanted us to address the comment.

---

## [Decision Letter · Decision Letter 2]

21 Nov 2023

Coming of Age in Recovery: The prevalence and correlates of substance use recovery status among adolescents and emerging adults

PONE-D-23-08858R2

Dear Dr. Smith,

We’re pleased to inform you that your manuscript has been judged scientifically suitable for publication and will be formally accepted for publication once it meets all outstanding technical requirements.

Kind regards,

Ricardo de Mattos Russo Rafael, Ph.D.

Academic Editor

PLOS ONE

Reviewers' comments:

Reviewer's Responses to Questions

**Comments to the Author**

1. If the authors have adequately addressed your comments raised in a previous round of review and you feel that this manuscript is now acceptable for publication, you may indicate that here to bypass the “Comments to the Author” section, enter your conflict of interest statement in the “Confidential to Editor” section, and submit your "Accept" recommendation.

Reviewer #2: All comments have been addressed

2. Is the manuscript technically sound, and do the data support the conclusions?

Reviewer #2: Yes

3. Has the statistical analysis been performed appropriately and rigorously? 

Reviewer #2: Yes

4. Have the authors made all data underlying the findings in their manuscript fully available?

Reviewer #2: No

5. Is the manuscript presented in an intelligible fashion and written in standard English?

Reviewer #2: Yes

6. Review Comments to the Author

Reviewer #2: Thank you for the in-depth response to my suggestion. I agree with the argumentation, and I wish to the authors many success following the publication of this study.

7. PLOS authors have the option to publish the peer review history of their article (what does this mean?). If published, this will include your full peer review and any attached files.

Reviewer #2: No

---

## [Editor Report · Acceptance letter]

6 Dec 2023

PONE-D-23-08858R2 

Coming of Age in Recovery: The prevalence and correlates of substance use recovery status among adolescents and emerging adults 

Dear Dr. Smith:

I'm pleased to inform you that your manuscript has been deemed suitable for publication in PLOS ONE. Congratulations! Your manuscript is now with our production department. 

Kind regards, 

on behalf of

Dr. Ricardo de Mattos Russo Rafael 

Academic Editor

PLOS ONE